# Economic Burden of Not Complying with Canadian Food Recommendations in 2018

**DOI:** 10.3390/nu11102529

**Published:** 2019-10-20

**Authors:** Olivia K. Loewen, John P. Ekwaru, Arto Ohinmmaa, Paul J. Veugelers

**Affiliations:** Population Health Research Intervention Unit, School of Public Health, University of Alberta, Edmonton, AL T6G 2T4, Canada; oloewen@ualberta.ca (O.K.L.); ekwaru@ualberta.ca (J.P.E.);

**Keywords:** health economics, health care costs, food consumption, dietary assessment, national survey, Canada, CCHS

## Abstract

Poor diet has been identified as a major cause of chronic disease. In this study we estimated the 2018 economic burden of chronic disease attributable to not complying with Canadian food recommendations. We retrieved the chronic disease risk estimates for intakes of both protective (fruit, vegetables, milk, whole grains, nuts and seeds) and harmful foods (sugar-sweetened beverages, processed meat, red meat) from the Global Burden of Disease Study, and food intakes from the 2015 Canadian Community Health Survey 24-hour dietary recalls (*n* = 19,797). Population attributable fractions (PAFs) were calculated for all food–chronic disease combinations, and mathematically adjusted to estimate the 2018 annual direct (hospital, physician, drug) and indirect (human capital approach) economic burden for each disease. Not meeting the eight food recommendations was estimated to be responsible for CAD$15.8 billion/year in direct (CAD$5.9 billion) and indirect (CAD$9.9 billion) costs. The economic burden of Canadians under-consuming healthful foods exceeded the burden of overconsumption of harmful foods (CAD$12.5 billion vs. CAD$3.3 billion). Our findings suggest poor diet represents a substantial economic burden in Canada. Interventions may be more effective if they are wide in focus and promote decreased consumption of harmful foods alongside increased consumption of healthful foods, with emphasis on whole grains and nuts and seeds.

## 1. Introduction

Noncommunicable chronic diseases such as cancers, cardiovascular disease, and type 2 diabetes, remain the leading cause of death in Canada [1]. The direct (i.e., hospitalization, physician, and drug costs) and indirect (i.e., lost productivity, morbidity, and premature mortality) health care costs associated with chronic disease amount to billions of dollars each year according to the Economic Burden of Illness (EBIC) study [2]. Extensive research has identified poor diet quality as a major, though modifiable, cause of chronic disease [3]. In Canada, poor diet represents a substantial contributor to the burden of chronic disease, with only smoking ranking higher [4]. Given the important role of poor diet in chronic disease, the current economic burden in terms of direct and indirect costs of unhealthy eating in Canada needs to be characterized in order to identify the potential economic benefits of health promotion strategies targeting Canadians’ diets.

Various approaches have been utilized to measure the burden of chronic disease due to suboptimal diet. Recently, the Global Burden of Disease Diet Collaborators released a systematic analysis of the health effects of dietary risks in 195 countries [5]. They reported that 11 million deaths from noncommunicable diseases in 2017 were attributable to dietary factors, with low intake of whole grains, vegetables, and fruit and high intake of sodium as the leading dietary risk factors for deaths globally [5].The most common approach to estimating the economic burden of unhealthy eating is to quantify the impact of a single dietary factor, such as inadequate consumption of vegetables and fruit [6,7], dairy [8,9], meat [10], and fiber [11], and overconsumption of sugar-sweetened beverages [12,13]. However, these approaches do not capture the full picture and underestimate the overall economic burden of unhealthy eating.

Fewer studies have estimated the overall economic burden of poor diet quality or unhealthy eating. The existing studies vary as to whether they consider specific dietary intakes, and use an incidence-based (estimate of the lifetime costs of a condition from onset until recovery or death) or prevalence-based approach (estimate costs of a condition in a set period of time (e.g., one year), regardless of onset) to estimate the economic burden. A study of five European countries (France, Italy, Germany, Spain, UK) used an incidence-based approach to calculate the economic burden of unhealthy diets on type 2 diabetes, projecting a total burden of €883 million (equivalent to CAD$1.2 billion) for the year 2020 [14]. In 2006, Scarborough et al. used a prevalence-based approach to estimate the economic burden of ill health (cardiovascular disease, cancer, type 2 diabetes, and dental caries) due to diet and other known risk factors, finding poor diet to be responsible for £5.8 billion per year in the U.K. (equivalent to CAD$9.5 billion) [15]. Using a prevalence-based approach for coronary heart disease, cancer, stroke, and diabetes, a study in the United States estimated an economic burden of USD$70.9 billion (equivalent to CAD$93 billion) attributed to poor eating patterns in 1995 [16]. In terms of Canadian estimates, Health Canada used a similar approach to estimate the direct and indirect costs of unhealthy eating in 1998 to be CAD$6.6 billion [17]. More recently, Nshimyumukiza et al. used a prevalence-based approach and the healthy eating index to determine the economic burden of eating a low-quality diet in 2004 and 2015 in Canada, finding a slight decrease in overall economic burden (CAD$13.21 to $13.08 billion), but an increase in burden in the elderly [18].

The World Health Organization (WHO) has identified Mediterranean and Nordic-style diets as evidence-based diets for the prevention and control of noncommunicable chronic diseases [19]. The Mediterranean diet is rich in vegetables, fruit, nuts, cereals, and olive oil and favors consumption of fish and poultry over red meat and processed meats. The Nordic diet is similar, with emphasis on vegetables, berries, whole grains, pulses, fatty fish, and rapeseed oil. As of 2018, the WHO reported 15 countries in the European region have policies or programs recommending adherence to these diets [19]. In the Canadian context, Health Canada has established food recommendations to help the public consume a balanced diet by promoting consumption of healthful foods such as fruit, vegetables, nuts and seeds, whole grains, and milk, while limiting intake of harmful foods such as red meat, processed meat, and sugar-sweetened beverages. However, these guidelines are not widely met in Canada [20]. Studying adherence to established recommendations yields direction to the work of both policy makers and practitioners. For the year 2014, we estimated the economic burden of chronic disease (cardiovascular disease, cancer, type 2 diabetes, chronic kidney disease) due to not meeting recommendations for healthful and harmful foods to be $13.8 billion using data from the 2004 Canadian Community Health Survey (CCHS) [21]. To our knowledge, outside of our previous study, no others have quantified the economic burden of chronic disease due to not meeting established food recommendations. Since the CCHS data were collected in 2004, there has been a shift in public perception of what constitutes an acceptable diet and more up to date dietary survey data are available from the 2015 CCHS. To support ongoing evaluation of existing nation-wide food policies and implementation of future dietary programs and interventions, a more recent estimation is needed. In this study we estimate the 2018 economic burden of chronic disease due to not complying with food recommendations in Canada using 2015 CCHS data.

## 2. Materials and Methods

We defined the economic burden of chronic disease for 2018 as that which could be potentially circumvented if all Canadians complied entirely with food recommendations. This analysis pertains to the difference in the number of servings of fruit, vegetables, whole grains, nuts and seeds, milk, red meat, processed meat, and sugar-sweetened beverages Canadians are recommended to consume, and what they actually consumed using representative, population-based data from 2015. The present analysis follows our previous approach that used 2004 dietary data to estimate the economic burden of unhealthy eating in Canada in 2014 [21] Our estimations involved a prevalence-based, bottom-up approach, which estimated the economic burden of chronic diseases attributable to foods for specific age and sex groups (for a set period of time, one year). The steps in this approach included: (1) selection of foods for inclusion and retrieval of dose–response relative risks (RRs); (2) analysis of diet history data; (3) calculation of PAFs; and (4) estimation of attributable direct and indirect health care costs. The four steps are outlined below. Full analytic details can be found in our previous analysis [21].

### 2.1. Food Selection

We selected foods for inclusion that were identified by the 2017 Global Burden of Disease Study (GBD) as having convincing or probable evidence against one or more chronic diseases, and for which recommendations are in effect [3]. We included the following eight foods: fruit (not including juice), whole grains, nonstarchy vegetables, fluid milk, nuts and seeds, processed meat, red meat, and sugar-sweetened beverages (SSB). The 2017 GBD also included legumes, however we did not include them in the analysis as Canadian recommendations for their consumption were not in effect at the time of the survey in 2015. We retrieved dose–response relative risks for each of the food–chronic diseases combinations from the 2017 GBD (with the exception of SSB, which were retrieved from the 2013 GBD [22] as the RR values in 2017 did not correspond to the conventional 226.8 g serving). Table 1 presents all included food–chronic disease combinations. All RRs used are presented in Appendix A.

### 2.2. Food Consumption Data Analysis

We extracted data on Canadians’ food consumption from the CCHS Cycle 2.2 (Nutrition) 2015, a cross-sectional national survey conducted by Statistics Canada in 2015. Respondents (*n* = 20,487) completed a 24-hour dietary recall, and of those respondents, 7623 (37%) completed a second recall. Details on sampling and data collection methodology are available elsewhere [23]. We excluded respondents that were <2 years of age, records where only breast milk or no foods were recorded, and records marked as invalid. In total, we included 24-hour dietary recall records from 19,797 respondents.

Established recommendations from the Canadian government outline a dietary pattern for Canadians to help meet nutrient requirements and prevent chronic diseases. In 2019, a new Canadian food guide was released, however Eating Well with Canada’s Food Guide (CFG 2007) was used in the present analysis, as it was in effect at the time of the survey [24]. Specific recommendations for four food groups (grain products, milk and alternatives, vegetables and fruits, meat and alternatives) and other dietary factors (water, fats, and oils) are outlined for each age and sex group. As all foods in our analysis (with the exception of SSB) are part of CFG food groups, we defined the consumption of these foods in CFG serving sizes.

To calculate the number of CFG servings from the 24-hour dietary recalls, we used the Canada Food Guide file, which categorizes all included foods according to Health Canada’s Canadian Nutrient File (CNF/CFG) tool [25,26]. The tool classifies foods into the four CFG groups and further subgroups. Within those subgroups, foods are tiered based on quantities of sodium, fat, and sugar. Foods in Tiers 1–3 count towards CFG servings, and foods in Tier 4 (higher in sugars, fat, and sodium) are generally not counted [26].

Table 1, column 1, presents the food groups included in our analyses. We included all foods belonging to each of the CFG groups, with a few exceptions. We excluded potatoes, corn, and fortified soy beverages from the relevant subgroups [27]. For red meat, only meat from the muscle of mammals (pork, beef, veal, lamb) that met the International Agency for Research on Cancer’s red meat definition were included [28]. For healthful foods, only those in Tiers 1–3 were included. For harmful foods, we included foods in Tier 1–4 [26]. Beverages in the CNF/CFG beverage subgroup with ≥50 kcal/226.8 g were included and considered as SSBs [27]. We assumed one serving of SSB was equal to 226.8 g (Table 1).

We assumed that complying with food recommendations was associated with the lowest disease risk. As CFG does not contain separate fruit and vegetable recommendations (only a combined recommendation), we estimated these using the GBD 2015 Theoretical Minimum Risk Exposure (TMRE). Vegetable and fruit servings were multiplied by ~0.63 and ~0.37 for vegetables and fruit, respectively [22]. As CFG age categories do not coincide with those in the EBIC economic cost calculations, we assumed the following: for individuals in the ≤14 years EBIC age category we used CFG recommendations for children 4–8 years, as this was the nearest to the middle of the EBIC category; for adult age groups, we used the CFG recommendation that applied to the majority of each EBIC age and sex group. As CFG does not have specific recommendations for consumption of processed meat, red meat, or nuts and seeds, we used established recommendations from other reliable Canadian organizations. For processed meat and red meat, Canadian Cancer Society recommendations were used (processed meat: only for special occasions, assumed <0.05 servings/day based on 2015 GBD TMRE; red meat: ≤385 g cooked servings/week) [29]. For nuts and seeds, the recommendation of 30 g/day (~1 CFG serving of nuts and seeds) from the 2016 Canadian Cardiovascular Society Guidelines [30] was used. For SSB, a recommendation of 5g/day was used, based on the 2015 GBD TMRE. Recommendations are summarized in Table 1.

We used the National Cancer Institute (NCI) method to estimate the usual intake distributions for each age and sex group [31]. Details on the method can be found elsewhere [31]. Similar to our previous study and others [21,32,33,34], we stratified the sample into children (≤14 years) and adults (≥15 years). However, because of a smaller sample size in 2015 CCHS compared to 2004, we were unable to run sex-stratified adult models, and instead used dummy variables. We used the model for food consumed non-episodically (i.e., consumed by almost everyone) for all foods except vegetables and milk, where a model for foods consumed episodically (i.e., not consumed daily by everyone) was used. For nuts and seeds models, only one-day intakes were used as few respondents consumed this food.

Using the NCI method, we estimated the number of servings of each food consumed by each EBIC age and sex group. Information on the proportion of the population consuming half CFG serving increments for all foods were obtained, with the exception of SSBs, where full serving increments were obtained. For foods with recommended intake near zero, we assumed consumption below the following cut-points was not associated with elevated risk: 56.7 g/day of SSB, 0.25 serving/day of processed meat, as accurate estimates below these cut-points could not be calculated.

### 2.3. Population Attributable Fraction Calculations

Similar to our previous approach, we calculated population attributable fractions (PAF) using dose–response RRs and food consumption data. PAFs estimate the fraction of disease cases that would be prevented if the exposure were to be eliminated from the population (i.e., if everyone complied with food recommendations) [35]. The standard PAF equation is: P(RR-1)/(P(RR-1) + 1), where P is the risk factor prevalence and RR is the relative risk of disease. However, as the extent to which each recommendation is met influences the disease risk, we used the method described by Krueger et al. to account for multiple risk exposure levels [35]. Further explanation of the equation can be found in our previous analysis [21].

Similar to our previous analysis, we calculated separate PAFs for each food and chronic disease combination by sex for each of the following age groups: ≤14 years, 15–34 years, 35–54 years, 55–64 years, 65–74 years, and ≥75 years. PAF values were then combined for each chronic disease. To avoid double counting, we used the following approach suggested by Krueger: combined PAF = 1-((1-PAF_1_)(1-PAF_2_)…(1-PAF_n_)) [35]. This approach assumes the risk factors are independent and the joint effects of the risk factors are multiplicative. This is the same approach we used previously and has been used to combine dietary risk factors previously [3,36,37].

### 2.4. Estimation of Direct Health Care and Indirect Costs

Direct health care costs for each age and sex group, including hospital, physician, and drug costs, associated with each chronic disease were retrieved from the 2008 EBIC [2]. Costs were then multiplied by 2018 National Health Expenditure Trends (i.e., costs in 2018 Canadians dollars ($) [38] to estimate more current direct health care costs. We used the modified human capital approach as outlined by Krueger et al. to estimate indirect health care costs (including those associated with mortality, long-term disability, and short-term disability) [35]. In short, using the 1998 EBIC [39], a ratio of total indirect health care to direct costs was calculated for each disease [21]. The calculated ratios were then multiplied by the 2018 direct costs to obtain indirect costs for each chronic disease by age and sex group. The direct and indirect costs were then multiplied by the relevant PAF to determine the costs attributable to unhealthy eating. To determine the economic burden attributable to individual foods, a disintegration step was applied [35,40]. All costs are reported in 2018 Canadian dollars ($).

Similar to Krueger et al. [41], a sensitivity analysis was conducted to estimate the upper and lower bounds of our cost estimates. To do this, we recalculated the PAFs using the 95% confidence intervals for all included relative risks. We conducted a second sensitivity analysis excluding the relationships between SSBs and cancers (leukemia, thyroid, liver, pancreatic, breast, ovarian, uterine, and kidney), as unlike the 2013 GBD [42], the 2017 GBD [3] did not provide the respective relative risks (see discussion section).

## 3. Results

Food recommendation adherence in the Canadian population is summarized by age and sex groups in Table 2. Healthful food recommendations met least often include (range of percent adherence in brackets): nuts and seeds (0.5–6.3%), whole grains (1.3–5.3%), and vegetables (0.8–7.2%). In comparison, more Canadians complied with recommendations for fruit (5.4–25.1%) and milk (2.1–21.1%). On average, Canadians were more likely to meet recommendations for harmful foods, adhering to recommendations for processed meat (18.8–40.5%), sugar-sweetened beverages (18.4–36.2%), and red meat (21.7–86.5%). Additional information on the Canadian population’s consumption of specific numbers of food servings can be found in Appendix A.

Table 3 presents the combined PAF values for each disease. The combined PAF values were generally larger for males than females. The combined PAF values for diabetes and cardiovascular diseases (range: 14.5–94.4%) were generally higher than for chronic kidney disease and cancers (range: 0.1–24.9%). Additionally, combined PAF values for diabetes and cardiovascular diseases were generally larger for ages under 54 compared to 55 and over.

We estimated not meeting food recommendations for five healthful and three harmful foods to be responsible for a total economic burden of $15.8 billion (Table 4). Of this total, direct health care costs amounted to $5.9 billion and indirect costs constituted $9.9 billion of the burden. Table 4 is stratified by sex, and further separated by direct and indirect health care costs. The total economic burden of not meeting recommendations for each food is presented and further stratified by the diseases contributing to the burden. The summed total costs for each food are presented on the right of Table 4, with the total direct and indirect costs for males and females at the bottom of the table. Not complying with food recommendations accounted for approximately 3.93% of all hospital, physician, and drug costs in 2018. Men had over twice the costs of women, with $4.1 billion and $1.8 billion direct costs, and $6.8 billion and $3.1 billion indirect health care costs, respectively. Out of the economic burden attributable to poor diet, 58% of estimated health care costs were attributable to ischemic heart disease ($3.4 billion in direct health care costs, $5.7 billion indirect costs). Type 2 diabetes and colon/rectal cancer were also substantial contributors with $4.1 billion and $1.2 billion total direct and indirect costs, respectively. Fewer of the estimated costs were attributable to other cancers ($520 million), chronic kidney disease ($5.3 million), and stroke ($904 million). Using the 95% confidence intervals from the relative risks in our PAF calculations, we performed a sensitivity analysis and obtained the low and high estimates of the total economic burden cost to be $7.9 billion and $21.2 billion, respectively. The low to high estimate intervals for the direct and indirect costs were $3.0–7.8 billion and $4.9–13.5 billion, respectively.

The estimated economic burden of Canadians not consuming adequate amounts of recommended healthful foods (fruit, vegetables, nuts and seeds, whole grains, milk) greatly exceeded the economic burden of overconsumption of harmful foods (processed meat, red meat, SSBs) ($12.5 billion vs. $3.3 billion) (Table 4). Inadequate consumption of whole grains and nuts and seeds accounted for almost half of the economic burden attributable poor diet, with each accounting for over $1.4 billion and $2.3 billion in direct and indirect health care costs, respectively. Inadequate consumption of fruit and excess consumption of processed meat were each responsible for an estimated 14–16% of direct and indirect health care costs. Inadequate consumption of vegetables and excess consumption of SSBs were each, respectively, responsible for an estimated 10% and 4–6% of direct and indirect health care costs. Red meat and milk and were responsible for the lowest estimated economic burden at <5% (Table 4). The proportion of the economic burden attributable to each food is summarized in Figure 1.

Table 5 presents the estimated direct and indirect economic burden attributable to poor diet by age and sex groups. Men were responsible for 66.9% of the economic burden ($10.6 billion) and men in age groups 35–54 and 55–64 were responsible for the largest proportion of the economic burden at 25.7% and 27.2%, respectively. Among women, those aged 65–74 and 75 and over were responsible for the largest proportion of the economic burden at 25.2% and 29.8%, respectively. Overall, men aged 55–64 alone were responsible for the largest proportion of the total burden at 18.2% (Table 5).

## 4. Discussion

We estimated that a total of $15.8 billion in direct and indirect health care costs could potentially be avoided if Canadians fully complied with established food recommendations. Our sensitivity analysis revealed this economic burden could be as low as $7.9 billion or high as $21.2 billion. Out of the eight foods considered, inadequate consumption of whole grains and nuts and seeds were the largest contributors, together accounting for over half of the burden.

Our estimates reinforce diet as one of the larger contributors to the economic burden of chronic disease, alongside excess bodyweight, tobacco smoking, and physical inactivity. In 2012, the economic burden for each of these latter risk factors were estimated to be $23.5 billion, $19.5 billion, and $10.6 billion, respectively [41]. These estimates also used PAFs and similar cost estimation methods, however they took a broader approach to costs and included those of other health care expenditures, other health professionals, and health research, in addition to the hospital care, physician, and drug costs for each disease. If we had included all the costs considered by Kruger et al. in our present study, our estimate could be 30% to 50% higher ($20.5–23.7 billion) [41]. Nevertheless, our estimate of $15.8 billion positions unhealthy diet after the burden of smoking and higher than the burden of physical inactivity. The upper end of our sensitivity analysis ($21.2 billion) would position the burden of unhealthy eating just after that of excess weight and even more costly than the economic burden of smoking.

Putting the estimates for the economic burden of poor diets in Canada in perspective, the most recent estimate before ours was conducted over 20 years ago and revealed the costs to be $6.6 billion in 1998. Our previous estimate of $13.8 billion using 2004 data and our current estimate of $15.8 billion using 2015 data and 2018 costs, depict the economic burden of chronic disease attributable to poor diet as steadily increasing. On the contrary, in Nshimyumukiza et al.’s [18] analysis of the economic burden of poor diet quality in terms of the Healthy Eating Index, the temporal changes in Canadians’ diets from 2004 to 2015 were found to improve, translating to a $133 million decrease in economic burden ($13.21 billion vs. $13.08 billion). This study noted that the economic burden decreased among males, but increased among females ($333 million) and among those over 65 years ($200 million). In the present study, the proportion of men’s costs compared to women’s costs showed marginal improvements when compared to our previous estimates (66.9% vs. 67.3%). However, men aged 55–64 remained the most burdensome, accounting for 18.2% of the total economic burden. The elderly (men and women over the age of 65 combined) were responsible for 48% of the total costs in the present analysis, a slight increase from our previous analysis, where they were responsible for 46% of the burden. In general, the proportion of costs between age and sex groups remained substantially similar over time.

The $3 billion (22%) increase from our estimate in 2014 to the present estimate appears dramatic. When we eliminate inflation and the general increase in health care costs over the last four years by using the 2014 National Health Expenditure Trends in the present models [38], the economic burden is estimated to be $13.89 billion, an increase of only $90 million from our previous estimate. Additionally, discrepancies between the 2004 and 2015 CCHS data have come to light, with a 250 kcal lower energy intake across the entire 2015 sample compared to 2004 [44]. This lower energy intake was consistent across all categories of respondents for many foods, including SSBs, meat, milk, and cereal grains. However, energy intake was higher for nuts [44]. As our analysis depends on both consuming enough of certain foods while limiting others, the effect of underreporting (on all foods except nuts) has bidirectional effects on our estimate and complicates comparisons between studies. The same issues in comparison between the 2004 and 2015 CCHS may exist in Nshimyumukiza et al.’s analysis, and it is not known if a 250 kcal decrease in 2015 influenced the apparent improvement in diet quality. However, the assessment of diet quality in terms of the HEI score may have circumvented inaccuracies in the 2015 CCHS estimation of quantity. Comparison issues aside, this study also found the majority of Canadians have a poor-quality diet that results in an attributable high economic burden, and that there is little evidence of substantial change between 2004 and 2015.

We revealed that the highest proportion of the economic burden resulted from inadequate intake of nuts and seeds, whole grains, and fruit. These proportions appeared similar to those based on the 2004 CCHS data [20]. In contrast, in the present analysis using the 2015 data (vs. using the 2004 data), vegetables (10.9% vs. 8.5%) and SSBs (6.3% vs. 5.2%) had higher proportions of attributable burden, while costs due to red meat consumption decreased slightly (1.5% vs. 2.9%). The total costs of processed meat and milk remained proportionally similar. Concordant with Nshimyumukiza et al.’s temporal analysis of diet quality in Canada, our analysis reveals little evidence of substantial change in Canadians’ diets [18].

It is possible that inadequate consumption of whole grains and the associated increase in economic burden could be due to the rise of gluten-free and low-carbohydrate diets in the past decade [45,46]. Gluten is a protein found in a variety of grains, including wheat, rye, and barley. Some individuals have an intolerance to gluten and individuals with Celiac disease have an autoimmune reaction and can experience severe side effects if they consume the protein. In Canada, approximately 1% of the population has Celiac disease and an additional 1–6% report gluten sensitivity, however the true prevalence is difficult to establish as non-Celiac gluten sensitivity is often self-diagnosed [45,47]. An estimated 25% of American consumers reported consuming gluten-free food in 2015, and it is likely Canadian consumption of gluten free food was similar [48]. It is still possible to meet whole grain recommendations by consuming non-gluten whole grains such as buckwheat, millet, and quinoa, however those individuals on low carbohydrate diets are unlikely to consume these carbohydrate-containing items. Thus, the popularity of gluten-free and low-carbohydrate diets in Canada is likely responsible for the, on average, lower consumption of whole grains and for an increase in economic burden.

The present analysis used the 2017 GBD dietary- and disease-relative risks for all food–disease combinations except for SSBs. The 2013 GBD dietary risks were used instead for SSBs, as the 2017 risks are no longer presented in standard serving units (226.8 g/day). To reflect that unlike the 2013 GBD, the 2017 report does not present risks for SSBs and cancers, a sensitivity analysis was conducted to estimate the cost when the cost of these cancers are not included. We found these cancers only account for 4% ($39.5 million) of the total economic burden of excess consumption of SSBs ($996 million).

Our study suggests that investment in promotion of healthy eating has the potential of substantial savings in direct and indirect health care costs. The present analysis provides information on economic benefits of what is to gain if Canadians were to eat according to established recommendations, and gives direction and guidance to health promotion. We revealed which specific food group yielded the biggest economic burden; these food groups, whole grains, nuts and seeds, and fruit, should receive priority. We also revealed that 79.2% of the economic burden is caused by underconsumption of healthful foods and 20.8% by overconsumption of harmful foods, suggesting that promotion of healthful foods, rather than restricting or taxing harmful foods, has the bigger potential to reduce the risk for chronic disease and associated economic burden. We further identified those age and sex groups wherein improving their diets could yield the greatest improvement in chronic disease prevention and reduction in economic burden. Whereas it is clear that investments in effective health promotion may result in enormous cost savings, it is not the only sector that needs attention [6]. Strategizing food production is essential to ensure the availability of healthy foods [6]. The latter issue is not uncomplicated in light of the demands for agricultural lands to ensure sustainable food production to meet the Western food recommendations [49]. 

There are several strengths in the present study. We used 24-hour dietary recalls from a large representative sample of the Canadian population. Only those foods identified by the GBD to cause chronic disease were included. We considered multiple dietary components, however our estimation was limited by not including the costs associated with over- and underconsumption of nutrients (e.g., fiber, sodium, fat). Established limitations of dietary research apply to the present study: assessment of dietary intake relies on self-report, which is susceptible to error (e.g., over-reporting of healthy foods and under-reporting of unhealthy foods), and reporting of food frequency does not directly translate into absolute consumption. In addition, we assumed the costs by disease and the ratio of indirect to direct costs did not change over time. However, this assumption has been made in other similar studies [6,7,40,50]. Our results do not estimate the lifetime costs of not complying with established food recommendations, a limitation of all prevalence-based studies. The use of an incidence-based approach would have allowed for the estimate of lifetime costs. However, given that many of the chronic diseases included have a long duration, and would require a long follow-up, the prevalence-based approach was appropriate. Lastly, our characterization of the economic burden may represent an underestimation, as we limited our direct cost estimation to hospital, physician, and drug costs and did not include costs related to other health care expenditures and health professionals, which have been included in some previous Canadian studies [7,41].

## 5. Conclusions

The economic burden due to unhealthy eating has remained very high over the last 11 years, despite the health promotion efforts to improve diet quality. Relatively small dietary changes at the population level can amount to substantial cost savings. Policy and decision makers are encouraged to develop more effective targeted nutritional programs to limit consumption of harmful foods, and in particular, promote consumption of healthful foods such as whole grains and nuts and seeds.

## Figures and Tables

**Figure 1 nutrients-11-02529-f001:**
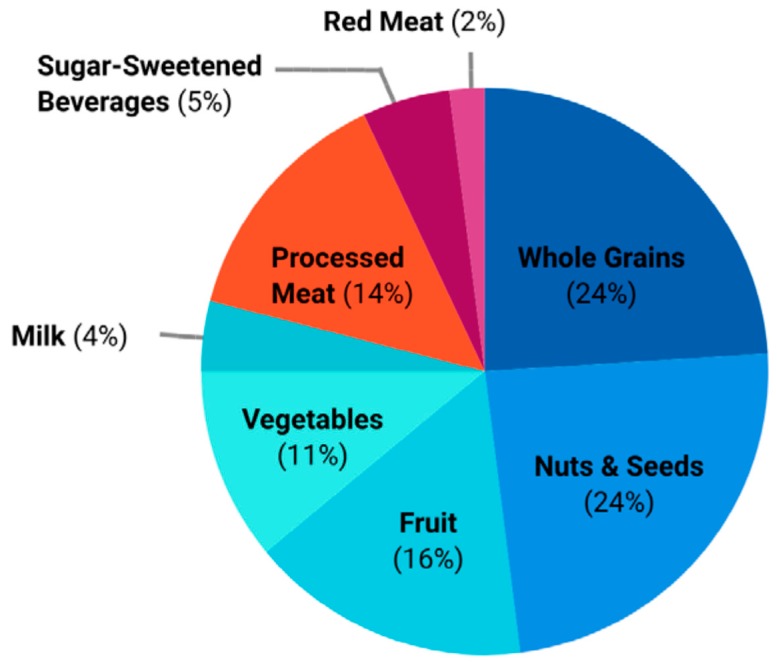
Proportion of CAD$15.8 billion economic burden attributable to not meeting established food recommendations.

**Table 1 nutrients-11-02529-t001:** Foods and chronic diseases included in estimation of the economic burden of unhealthy eating in Canada.

Food	Chronic Disease	Food Recommendation	Serving Size	CNF/CFG Group Subgroup and Tiers Included for Analysis
Fruit	Cancer: Mouth; Larynx; Esophagus; Trachea, Lung, and Bronchus Cardiovascular Disease: Ischemic Heart Disease; Ischemic Stroke; Hemorrhagic Stroke Diabetes	Female and Male ^†^ ≤14 years: 2 servings/day 15+ years: 3 servings/day	80 g	Vegetables and Fruit ⚬ Fruit other than juice (Tiers 1–3; subgroup codes: 1121, 1122, 1123)
Vegetables	Cardiovascular Disease: Ischemic Heart Disease; Ischemic Stroke; Hemorrhagic Stroke	Female ^†^ ≤14 years: 3 servings/day 15+ years: 4 servings/day Male ^†^ ≤14 years: 3 servings/day15–54 years: 5 servings/day 55+ years: 4 servings/day	80 g	Vegetables and Fruit (note all corn and potatoes were removed from relevant categories): ⚬ Vegetables, dark green (Tiers 1–3; subgroup codes 1211, 1212, 1213) ⚬ Vegetables, deep yellow or orange (Tiers 1–3; subgroup codes 1221, 1222, 1223) ⚬ Vegetables, other (Tiers 1–3; subgroup codes 1241, 1242, 1244)
Whole Grains	Cardiovascular Disease: Ischemic Heart Disease; Ischemic Stroke; Hemorrhagic Stroke Diabetes	Female ^††^ ≤14 years: 2.5 servings/day 15+ years: 3 servings/day Male ^††^ ≤14 years: 2.5 servings/day 15–54 years: 4 servings/day 55+ years: 3.5 servings/day	35 g	Grain Products ⚬ Whole grain (Tiers 1–3; subgroup codes 2101, 2102, 2103)
Nuts and Seeds	Cardiovascular Disease: Ischemic Heart Disease Diabetes	30 g (~1 Canada Food Guide serving)/d^*^	30 g	Meat and Alternatives ⚬ Nuts and seeds (Tiers 1–3; subgroup codes 4601, 4602, 4603)
Milk	Cancer: Colorectal	2 cups/day ^††^	1 cup (~257.8 g)	Milk and Alternatives (all soy beverages removed) ⚬ Fluid milk and fortified soy-based beverages (Tiers 1–3; subgroup codes: 3101, 3102, 3103)
Red Meat	Cancer: Colorectal Diabetes	No more than 3 × 85 g servings/week (rounded this to <0.5 servings/day) **	75 g	Meat and Alternatives (all offal and meat not meeting the International Agency for Research on Cancer criteria for red meat were excluded): ⚬ Beef, game and organ meats (Tiers 1–4; subgroup codes 4101, 4102, 4103, 4104) ⚬ Other meats (pork, veal, lamb) (Tiers 1–4; subgroup codes 4201, 4202, 4203, 4204)
Processed Meat	Cancer: Colorectal Cardiovascular Disease: Ischemic Heart Disease Diabetes	Only for special occasions (assumed 0.05 servings/day servings/day) ** Assumed those who usually consumed <0.25 servings/day had no elevated risk of disease	75 g	Meat and Alternatives ⚬ Processed meats (Tiers 1–4; subgroup codes 4801, 4802, 4803, 4804)
Sugar-Sweetened Beverages	Cancer: Esophagus; Thyroid; Liver; Pancreas; Colorectal; Breast (post-menopausal); Ovary, Uterus; Kidney; Leukemia Cardiovascular Disease: Ischemic Heart Disease; Ischemic Stroke; Hemorrhagic Stroke Diabetes Chronic Kidney Disease	5 g g/day *** Assumed those who usually consumed <56.7 g/day had no elevated risk of disease.	226.8 g	Beverages sweetened with sugar with ≥50 kcal/226.8 g were included from the following categories ⚬ Subgroup codes 5410 and 5420

CNF: Canadian Nutrient File, CFG: Canada’s Food Guide. ^†^ Based on recommendations from Canada’s Food Guide and the 2017 GBD; ^††^ based on Canada’s Food Guide recommendations; * based on 2016 Canadian Cardiovascular Society Guidelines for the Management of Dyslipidemia for the Prevention of Cardiovascular Disease in the Adult; ** based on Canadian Cancer Society recommendations; *** based on GBD 2013 recommendations.

**Table 2 nutrients-11-02529-t002:** Food recommendation adherence percentages of 2015 Canadian population stratified by age and sex.

	Canadian Population in 2018 (‘000) ^†^	Healthful Foods (% Consuming at or Above Recommendation)					Harmful Foods (% Consuming at or Below Recommendation)		
		Nuts and Seeds	Whole Grains	Fruit	Vegetables	Milk	Processed Meat ^††^	Sugar-Sweetened Beverages *	Red Meat
**Females**									
≤14 years	2780.6	1.1	4.8	19.0	15.9	27.6	49.3	15.8	70.1
15–34 years	4703.0	1.5	1.6	5.8	5.8	9.7	67.0	24.6	59.2
35–54 years	4989.7	3.1	1.6	8.7	13.4	6.0	70.6	48.7	44.2
55–64 years	2378.0	4.3	2.4	8.9	15.7	4.7	78.4	57.7	42.0
65–74 years	1629.2	1.9	1.5	9.0	13.1	5.5	73.2	61.7	48.0
75+ years	1434.6	1.5	2.2	8.1	6.7	6.9	84.0	66.7	49.8
**Males**									
≤14 years	2928.0	0.8	9.1	15.7	12.2	37.8	34.4	12.5	54.8
15–34 years	4825.4	2.5	1.3	5.2	7.0	16.8	22.4	9.9	22.4
35–54 years	5002.8	3.5	1.8	9.9	9.4	6.7	28.3	29.3	17.1
55–64 years	2347.5	3.5	3.5	12.6	26.1	5.8	26.2	42.5	22.1
65–74 years	1516.3	3.0	3.1	11.6	21.3	7.1	38.1	54.4	27.9
75+ years	1005.1	2.8	2.8	9.4	17.4	11.8	51.2	68.9	36.3

^†^ Statistics Canada 2018 [43], ^††^ ≤0.25 servings/day; * <56.7 g/day.

**Table 3 nutrients-11-02529-t003:** Combined population attributable fractions for foods with established food recommendations presented by age, sex, and chronic disease.

	Females						Males						Foods Included in Calculations
	≤14 years	15–34 years	35–54 years	55–64 years	65–74 years	75+ years	≤14 years	15–34 years	35–54 years	55–64 years	65–74 years	75+ years	
**Cancer**													
Mouth	2.3	5.8	5.5	5.2	5.3	5.4	2.4	6.1	5.7	5.5	5.6	5.6	F
Larynx	2.3	5.8	5.5	5.2	5.3	5.4	2.4	6.1	5.7	5.5	5.6	5.6	F
Thyroid	0.3	0.3	0.2	0.2	0.1	0.1	0.5	0.5	0.5	0.4	0.3	0.2	SSB
Trachea, Lung, and Bronchus	4.2	10.1	9.6	9.1	9.3	9.5	4.3	10.6	10.0	9.6	9.7	9.9	F
Esophagus	8.5	19.3	18.3	17.5	17.8	17.9	8.9	20.5	19.3	18.5	18.6	18.7	F, SSB
Liver	0.3	0.4	0.2	0.2	0.2	0.1	0.7	0.8	0.5	0.5	0.4	0.3	SSB
Pancreas	0.1	0.1	0.1	0.1	0.1	0.1	0.2	0.2	0.2	0.1	0.1	0.1	SSB
Colorectal	15.4	18.5	19.4	19.8	19.0	18.5	16.9	23.5	23.9	24.9	23.2	22.6	M, RM, PM, SSB
Kidney	0.5	0.6	0.4	0.3	0.3	0.2	0.6	0.7	0.5	0.4	0.3	0.2	SSB
Leukemia	0.3	0.3	0.2	0.2	0.1	0.1	0.2	0.2	0.2	0.2	0.1	0.1	SSB
Post-Menopausal Breast				0.1	0.1	0.1							SSB
Ovary	0.1	0.1	0.0	0.0	0.0	0.0							SSB
Uterus	1.0	1.0	0.7	0.6	0.5	0.3							SSB
**Cardiovascular Diseases**													
Ischemic Heart Disease	86.2	86.8	64.1	54.1	47.2	41.7	88.3	93.0	73.9	60.5	53.1	46.6	F, V, WG, NS, PM, SSB
Ischemic Stroke	76.4	89.9	61.9	47.3	34.0	14.5	76.5	94.4	70.5	50.5	36.2	15.5	F, V, WG, SSB
Hemorrhagic Stroke	62.9	79.5	56.4	46.8	37.0	17.1	63.2	86.3	64.8	49.9	39.3	18.3	F, V, WG, SSB
**Diabetes**	64.1	70.4	54.8	45.9	37.3	23.6	71.5	83.2	68.4	57.9	47.5	30.6	F, WG, NS, RM, PM, SSB
**Chronic Kidney Disease**	1.2	1.2	0.8	0.7	0.5	0.2	1.4	1.6	1.1	1.0	0.7	0.3	SSB

F = fruit; V = vegetables; M = milk; WG = whole grains; NS = nuts and seeds; RM = red meat; PM = processed meat; SSB = sugar-sweetened beverages.

**Table 4 nutrients-11-02529-t004:** Economic burden in Canada in 2018 of the eight foods that have established recommendations.

	2018 Costs ($CAN)						
	Females		Males		Females and Males		
	Direct Costs	Indirect Costs	Direct Costs	Indirect Costs	Direct Costs	Indirect Costs	Total Direct Health and Indirect Costs
**Whole Grains**	479,696,161	732,962,516	1,013,846,354	1,592,873,190	1,493,542,514	2,325,835,706	3,819,378,221
Ischemic Heart Disease	231,644,786	404,966,223	608,802,593	1,064,321,328	840,447,379	1,469,287,550	2,309,734,929
Ischemic Stroke	28,258,369	49,401,867	41,805,494	73,085,232	70,063,862	122,487,098	192,550,961
Hemorrhagic Stroke	22,328,761	39,035,604	27,651,485	48,340,901	49,980,247	87,376,505	137,356,752
Diabetes	197,464,245	239,558,823	335,586,782	407,125,730	533,051,027	646,684,553	1,179,735,579
**Nuts and Seeds**	498,432,254	782,751,451	954,697,875	1,546,309,433	1,453,130,129	2,329,060,883	3,782,191,012
Ischemic Heart Disease	332,804,780	581,816,223	725,346,710	1,268,066,174	1,058,151,490	1,849,882,397	2,908,033,887
Diabetes	165,627,474	200,935,227	229,351,164	278,243,259	394,978,638	479,178,486	874,157,125
**Fruit**	326,305,644	596,430,274	576,656,369	1,049,843,530	902,962,012	1,646,273,804	2,549,235,816
Mouth Cancer	1,728,793	8,442,531	3,812,495	18,618,252	5,541,288	27,060,783	32,602,071
Laryngeal Cancer	476,108	2,325,065	2,075,830	10,137,280	2,551,938	12,462,345	15,014,283
Esophageal Cancer	2,936,399	14,339,852	9,816,589	47,939,145	12,752,988	62,278,997	75,031,985
Tracheal, Bronchial and Lung Cancer	21,104,161	103,061,806	23,602,668	115,263,226	44,706,829	218,325,032	263,031,861
Ischemic Heart Disease	137,370,373	240,153,738	317,238,513	554,602,952	454,608,886	794,756,690	1,249,365,576
Ischemic Stroke	32,067,357	56,060,819	40,893,335	71,490,577	72,960,692	127,551,396	200,512,088
Hemorrhagic Stroke	25,378,191	44,366,681	26,858,418	46,954,444	52,236,609	91,321,125	143,557,734
Diabetes	105,244,262	127,679,781	152,358,521	184,837,656	257,602,783	312,517,437	570,120,219
**Processed Meat**	146,720,962	240,898,174	693,784,426	1,158,558,074	840,505,388	1,399,456,248	2,239,961,636
Colon and Rectal Cancer	7,435,911	36,313,146	27,537,031	134,476,619	34,972,943	170,789,766	205,762,708
Ischemic Heart Disease	66,551,012	116,345,861	403,342,304	705,131,386	469,893,316	821,477,247	1,291,370,563
Diabetes	72,734,038	88,239,168	262,905,091	318,950,068	335,639,129	407,189,236	742,828,365
**Vegetables**	190,257,302	332,611,764	436,385,955	762,898,984	626,643,257	1,095,510,748	1,722,154,005
Ischemic Heart Disease	166,710,242	291,446,305	405,516,582	708,932,505	572,226,824	1,000,378,809	1,572,605,633
Ischemic Stroke	13,301,355	23,253,704	17,991,402	31,452,942	31,292,757	54,706,647	85,999,403
Hemorrhagic Stroke	10,245,706	17,911,755	12,877,971	22,513,537	23,123,677	40,425,292	63,548,969
**Sugar-Sweetened Beverages**	109,485,809	140,609,158	255,443,721	324,647,983	364,929,531	465,257,142	830,186,672
Esophageal Cancer	49,594	242,189	270,047	1,318,772	319,641	1,560,961	1,880,602
Thyroid Cancer	76,400	373,097	54,084	264,117	130,484	637,214	767,698
Liver Cancer	9898	48,336	80,200	391,655	90,098	439,991	530,089
Pancreatic Cancer	43,910	214,434	59,525	290,688	103,435	505,122	608,556
Colorectal Cancer	170,942	834,793	1,018,775	4,975,168	1,189,717	5,809,961	6,999,677
Breast Cancer	347,638	1,697,684			347,638	1,697,684	2,045,322
Ovarian Cancer	25,635	125,189			25,635	125,189	150,824
Uterine Cancer	380,325	1,857,312			380,325	1,857,312	2,237,637
Kidney Cancer	290,291	1,417,629	192,135	938,286	482,425	2,355,915	2,838,340
Leukemia	196,821	961,173	179,011	874,199	375,833	1,835,372	2,211,205
Ischemic Heart Disease	4,228,109	7,391,669	15,693,878	27,436,363	19,921,987	34,828,032	54,750,019
Ischemic Stroke	463,246	809,857	966,438	1,689,547	1,429,684	2,499,404	3,929,088
Hemorrhagic Stroke	697,652	1,219,649	1,089,408	1,904,525	1,787,059	3,124,174	4,911,233
Diabetes	101,401,444	123,017,768	234,020,470	283,907,948	335,421,915	406,925,716	742,347,631
Chronic Kidney Disease	1,103,904	398,379	1,819,751	656,715	2,923,655	1,055,094	3,978,750
**Milk**	50,108,728	244,705,113	63,055,698	307,931,418	113,164,426	552,636,531	665,800,957
Colon and Rectal Cancer	50,108,728	244,705,113	63,055,698	307,931,418	113,164,426	552,636,531	665,800,957
**Red Meat**	17,465,576	32,182,526	63,406,702	117,840,630	80,872,278	150,023,157	230,895,435
Colon and Rectal Cancer	2,995,312	14,627,554	11,148,157	54,441,833	14,143,469	69,069,386	83,212,855
Diabetes	14,470,264	17,554,973	52,258,546	63,398,798	66,728,810	80,953,770	147,682,580
**TOTAL COSTS**	1,818,472,435	3,103,150,976	4,057,277,100	6,860,903,243	5,875,749,535	9,964,054,219	15,839,803,754

**Table 5 nutrients-11-02529-t005:** Economic burden in Canada in 2018 by age and sex groups.

	2018 Costs ($CAN)			Percent of Costs	
	Direct Costs	Indirect Costs	Total Direct and Indirect Costs	Costs within Sex Group	Total Costs
**Females**	1,936,570,565	3,302,166,717	5,238,737,282	100.0%	33.1%
≤14 years	12,040,403	15,666,224	27,706,627	0.5%	0.2%
15–34 years	76,513,918	100,906,919	177,420,837	3.4%	1.1%
35–54 years	386,859,501	595,385,456	982,244,957	18.7%	6.2%
55–64 years	440,519,124	728,965,831	1,169,484,955	22.3%	7.4%
65–74 years	480,354,720	840,813,557	1,321,168,277	25.2%	8.3%
75+ years	540,282,899	1,020,428,729	1,560,711,628	29.8%	9.9%
**Males**	3,939,178,970	6,661,887,503	10,601,066,473	100.0%	66.9%
≤14 years	24,667,689.36	31,113,645.39	55,781,335	0.5%	0.4%
15–34 years	87,878,603	118,808,699	206,687,302	1.9%	1.3%
35–54 years	1,076,934,563	1,651,501,848	2,728,436,411	25.7%	17.2%
55–64 years	1,074,108,771	1,808,010,348	2,882,119,119	27.2%	18.2%
65–74 years	972,731,815	1,717,252,172	2,689,983,987	25.4%	17.0%
75+ years	702,857,528	1,335,200,790	2,038,058,318	19.2%	12.9%
All years	3,939,178,970	6,661,887,503	10,601,066,473	100.0%	66.9%

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
