# Peer review of "Economic Burden of Not Complying with Canadian Food Recommendations in 2018"

_nutrients, 2019, doi:10.3390/nu11102529_

Round 1

Reviewer 1 Report

The paper focuses on the Canadian food situation. That's ok, but the introduction should also consider more countries and their recommendations, e.g. Nordic diet or Mediterranean diet.

Table 4 is very confusing, better explained and possibly added to the appendix.

From my point of view, "marginal issues" should also be taken into account in the conclusion. For example, there was no mention at all of the way in which this study affects sustainability issues. Overall, the conclusion is not specific enough. We know that state diet programmes are not the only key. Food education in schools and also the idea of providing "healthy nutrition" with a coolness factor, here I would expect a few more ideas.

Author Response

Reviewer 1: The paper focuses on the Canadian food situation. That's ok, but the introduction should also consider more countries and their recommendations, e.g. Nordic diet or Mediterranean diet.
We thank the reviewer for this comment. We have expanded the introduction to make referral to various dietary recommendations and including the Mediterranean and Nordic diets as suggested by the reviewer.
The text now states: “The World Health Organization (WHO) has identified Mediterranean and Nordic-style diets as
evidence-based diets for the prevention and control of non-communicable chronic diseases.(19) The Mediterranean diet is rich in vegetables, fruit, nuts, cereals and olive oil and favors consumption of fish and poultry over red meat and processed meats. The Nordic diet is similar, with emphasis on vegetables, berries, whole grains, pulses, fatty fish, and rapeseed oil.”
Reviewer 1: Table 4 is very confusing, better explained and possibly added to the appendix.
Thank you to the reviewer for the feedback on Table 4. For clarity, we have explained the organization of Table 4 in the results text. For further clarity, we have added Figure 1 which provides a graphical representation of the key results from Table 4.
The text now states: “Table 4 is stratified by sex, and further separated by direct and indirect health care costs. The total economic burden of not meeting recommendations for each food is presented and further separated by the diseases contributing to the burden. The summed total costs for each food are presented on the right of Table 4, with the total direct and indirect costs for males and females at the bottom of the table…. The proportion of economic burden attributable to each food is summarized in Figure 1.”
Reviewer 1: From my point of view, "marginal issues" should also be taken into account in the conclusion. For example, there was no mention at all of the way in which this study affects sustainability issues.
We appreciate this suggestion: we have expanded the 8th paragraph of the Discussion section to put the implications of our findings in a broader context, including the sustainability issue mentioned by the reviewer. For example, we added: “Whereas it is clear that investments in effective health promotion my result in enormous cost savings, it is not the only sector that needs attention (6): Strategizing food production is essential to ensure the availability of healthy
foods (6). The latter issue is not uncomplicated in light of the demands for agricultural lands to ensure sustainable food production to meet the Western food recommendations.”
Reviewer 1: Overall, the conclusion is not specific enough. We know that state diet programmes are not the only key. Food education in schools and also the idea of providing "healthy nutrition" with a coolness factor, here I would expect a few more ideas.
We appreciate to be challenged to speak to the content of health promotion messages, in all modesty, we believe in this study we identified targets in terms of which foods to prioritize, which age and sex groups to focus on, etc. I believe we go beyond this study (and beyond our expertise) to speak to the content of health. We expanded the 8th paragraph of the Discussion to clarify this.

Reviewer 2 Report

Comments and suggestions for authors:

The manuscript titled “Economic burden of not complying with Canadian food recommendations in 2018” was overall a good attempt at quantifying the economic burden of chronic disease attributable to not meeting the Canadian food recommendations. Estimation of economic burden is an important aspect of healthy policy that should not be overlooked. Overall, this a good piece of writing. However, there are some issues that the authors need to address in the manuscript.

Comments:

In the Introduction, the authors mentioned that they had a previous study in a similar fashion with 2004 Canadian Community Health Survey data, but they did not mention how it is different from the current study other than the fact that there’s been a ‘shift in nation-wide policies and public perception’. I am a bit concerned with the novelty of the study so more details should be given regarding the rationale for the current study with updated data. Methods: sufficient details were given so that we could clearly follow the steps in the estimation process. Results: results are clearly organized but I believe they can be made more concise in terms of presentation. References: some spelling needs to be fixed (i.e., reference 3)

Author Response

Economic burden of not complying with Canadian Food Recommendations in 2018 -- by Olivia Loewen, Paul J. Veugelers et al.

Editor/Reviewer Comments

Author’s Response

Reviewer 2: In the Introduction, the authors mentioned that they had a previous study in a similar fashion with 2004 Canadian Community Health Survey data, but they did not mention how it is different from the current study other than the fact that there’s been a ‘shift in nation-wide policies and public perception’. I am a bit concerned with the novelty of the study so more details should be given regarding the rationale for the current study with updated data.

Thank you for providing us to opportunity to strengthen our study rational. As diet has a critical influence on the risk of chronic disease, updated estimates are necessary to provide feedback on existing food policies and inform future policy. We designed the present study to be substantially similar to our previous study to allow for comparison between 2004 and 2015.

The text now states: “To our knowledge, outside of our previous study, no others have quantified the economic burden of chronic disease due to not meeting established food recommendations. Since the CCHS data was collected in 2004, there has been a second survey in 2015 and a shift in public perception of what constitutes an acceptable diet. To support ongoing evaluation of existing nation-wide food policies and implementation of future dietary programs and interventions a more recent estimation is needed. In this study we estimate the 2018 economic burden of chronic disease due to not complying with food recommendations in Canada using 2015 CCHS data.”

Reviewer 2: Results: results are clearly organized but I believe they can be made more concise in terms of presentation.

Thank you for this suggestion. We have removed the redundancy between the results text and what is presented in the corresponding tables.

Reviewer 2: References: some spelling needs to be fixed (i.e., reference 3) 

Thank you for bringing this to our attention. We have fixed the spelling in this reference.